# Microbial Fertilization Improves Soil Health When Compared to Chemical Fumigation in Sweet Lily

**DOI:** 10.3390/jof8080847

**Published:** 2022-08-12

**Authors:** Hui Li, Hongyu Yang, Alejandro Calderón-Urrea, Yuanpeng Li, Lipeng Zhang, Yanlin Yu, Jiayi Ma, Guiying Shi

**Affiliations:** 1College of Horticulture, Gansu Agricultural University, Lanzhou 730070, China; 2College of Plant Protection, Gansu Agricultural University, Lanzhou 730070, China; 3Department of Biology, College of Science and Mathematics, California State University, Fresno, CA 90032, USA

**Keywords:** consecutive replant problems, metham-sodium, soil fumigation, fungi, bacterium, soil health, chemical hazard

## Abstract

Lanzhou Lily(*Lilium davidii)* var. *unicolor*, which is also known as sweet lily in China, is used as a type of food. This lily is distributed in narrow regions, propagates asexually, cultivates perennially, and cultivates commonly in serious consecutive replant problems (CRPs). Soil fumigation is commonly used to control soil-borne disease to alleviate crops’ consecutive replant problems (CRPs). However, due to the improper fumigation application, it is common to cause chemical hazard to crops. In this study, we designed a two-factor experiment to explore the bacterial and fungal community structure and some specific microbial groups in the lily rhizosphere soil after chemical versus bacterial fertilizer treatments, by using a metagenomic analysis of the treated soils. The results showed that metham-sodium soil fumigation (SMF treatment) significantly decreased plant growth, as well as it significantly decreased both soil fungal diversity and abundance at the OTUs levels, while Special 8™ microbial fertilizer supplement (MF treatment) significantly improved plant growth and increased fungal diversity and abundance. Under FM treatment, Chao1 richness and Shannon’s diversity increased by 6.70% and 35.09% compared to CK (no treatment). However, the bacterial diversity and abundance were not significantly changed among these treatments. The fungal and bacterial community structure were different in all treatments. In SMF treatment, the pathogenic fungal species *Fusarium oxysporum* increased compared to CK, but it significantly decreased in MF treatment; in MF and MMF treatments, some beneficial bacteria groups such as the bacterial phylum *Proteobacteria* and its member genus *Sphingomonas*, *as well as* the fungal genus *Mortierella,* increased compared to CK and SFM treatments, but the harmful bacterial genera *Gemmatimona* was decreased, as well as the harmful fungal genus *Cryptococcus*. Thus, we concluded that under chemical fumigation conditions, both fungal diversity loss and overall microorganism reduction, which impair multiple ecosystem function, in conjunction with the increase of harmful fungal species such as *Fusarium oxysporum,* are causes for soil degradation. On the other hand, under microbial fertilizer supplement, it was the fungal diversity increase, as well as these beneficial microorganisms groups’ accumulation, together with those harmful groups’ depletion, played important roles in restoring and improving soil health that suffered from the chemical fumigant hazard. In addition, the bacterial phylum *Proteobacteria* and its member genus *Sphingomonas* are involved in soil health recovery and promotion. The results also emphasized that whether soil is chemically fumigated or not, beneficial microorganism supplementary is effective in ensuring soil productivity.

## 1. Introduction

The large-scale degradation of farmland ecosystem and the loss of land productivity caused by consecutive replanting have attracted much attention [1]. Soil fumigation is a kind of soil disinfection technology, in which chemical fumigants are used to control plant disease, pests, and weeds by decomposing and producing toxic gas to kill harmful soil organisms in a closed space. However, in some cases, due to improper application, the incomplete decomposition of a fumigant commonly results in plant hazard. Therefore, it is of great importance to look for characteristics of soil contaminated with fumigants and the solution for restoring and improving soil health.

In recent years, great progress has been made in the study of functional microorganisms in farmland ecosystem due to the application of next-generation sequence (NGS) analysis. Among them, soil fungi are widely dispersed in terrestrial ecosystems and contribute a lot to ecosystem processes, including litter decomposition, nutrient cycling, and soil development [2,3]. Bacteria play important roles in maintaining soil nutrient cycling, biological control of soil-borne diseases as well as pests, and inducing plant disease resistance, etc., and its abundance and diversity are closely related to plant species, soil types, and cultivation techniques [4]. Some important microorganism communities were found to be related to soil health in continuous cropping systems. For example, among bacteria, the phylum *Acidobacteria* occur more frequently in the rhizosphere soil of healthy wheat plants than in diseased plants [5]. In addition, the genera *Lysobacter*, *Gemmatimonas,* and *Flavitalea* are related to apple continuous cropping disorder and the occurrence of tobacco bacterial wilt [6,7,8,9]. In addition, some fungal member genera, such as *Rhizoctonia Solani*, *Phytophthora* spp., *Cylindrocarpon* spp., and *Pythium* spp., are related to crops CRPs [10,11,12,13].

Lanzhou lily (*Lilium davidii* var. *unicolor*) is the only sweet lily in China, which is a type of famous local vegetable of Gansu province. It is an endemic species with a narrow distribution that is only suitable to the arid region, at the altitude of 2000–2600 m, in Gansu Province, western China. The lily is propagated asexually and cultivated perennially, so it is usually cultivated in a long-term continuous monoculture, which brings about serious CRPs. Our team found some beneficial microorganisms such as the bacterial phylum *Proteobacteria* and its member *Sphingomonas* and fungal genus *Penicillium* [14], and some harmful fungal genera such as *Acremonium*, *Fusarium*, *and Gibberella* can be associated with the lily CRPs [15].

In China, it is noted that using soil fumigation combined with microbial fertilizer keeps a soil microorganism community in balance and controls soil-borne disease by inducing the disease-resistant soil formation to solve consecutive replant problems (CRPs). In this soil management practice, soil chemical disinfection is carried out firstly, then microbial fertilizer is amended in the soil. Methlam-sodium is a kind of methyl derivative soil fumigant, which has many advantages (such as low toxicity, high efficiency, broad spectrum, and low cost etc.), and it is widely used to control crop disease and pest and weeding [16,17]. We carried out the experimental “soil fumigation (metham-sodium) + microbial fertilizer” to control this lily CRPs in 2017, but we were surprised to find out that in this cold and dry ecological area of northwest China, when metham-sodium was applied in spring, the lily was easily exposed to fumigants hazard, which inhabited the plantlets and bulb growth; however, microbial fertilizer could enhance the plant growth and alleviates CRPs and reduce the negative effects of metham-sodium fumigation hazard [18]. Due to this phenomenon, we speculated that the chemical substrates and microbial fertilizer must influence soil microorganism community impressively.

Fumigants have been used to control soil-borne pathogens of high-value crops for decades, and increasing evidence has indicated that they can alter bacterial abundance and diversity compared to non-treated soils [19]. In terms of previous research findings, we only viewed several literatures related to fumigant application on the fungal community structure change in soil, which differed from the amount of bacteria study [20,21]; thus, we know little about the effect of fumigants on non-target fungi. In addition, at present, many reports have focused on how microbial fertilizer effected soil microorganism community structures in normal conditions, while very few research reported on the soil microorganism community characters in chemical fumigation situation. Thus, it is necessary and interesting to understanding the potential fumigant impacts on soil fungal and bacterial community composition under fumigation and microbial fertilizer methods conditions.

In this study, we designed a two-factor experiment (one factor is soil metham-sodium fumigation that causes chemical hazard, and the other one is microbial fertilizer supplement, which is used to restore and improve soil health) and collected the fungal and bacterial community structure data in all treatments by using high-throughput sequencing technology. We found out that soil fumigation and microbial fertilizer changed the structure and diversity of microbial community in plant rhizosphere soil, and the fungal diversity enrichment and some special microorganism were involved in improving soil health recovery and promotion. To our excitement, we found out that the bacterial phylum *Proteobacteria* and its member *Sphingomonas* are involved in soil health recovery, which could be regarded as soil health indicators in Lanzhou Lily continuous cropping system, as we previously reported [14]. The objectives of this study are to determine whether the soil fumigation and microbial fertilizer changed the microorganism and what are the specific microorganism members detected. Similarly, we wanted to explore the specific fungal and bacterial members related to soil health damage, recovery, and promotion.

## 2. Materials and Methods

### 2.1. Experimental Design and Soil Sample Collection

The sample site was located in Jiangjiashan village, Lintao County, Gansu Province (western China, 103°53′12″~103°53′14″ E, 35°49′11″~35°49′13″ N, 2330 m elevation), where it is an arid zone in this loess Plateau. The soil is locally known as Huangmian soil (a deep soil layer, high water–storage capacity, pH 7.8, and organic matter 1.3%). Lanzhou lily has been cropped as food in this area for over 140 years.

We carried out the experiment on March 2019. The experimental field has replanted Lanzhou lily for 9 years (in 0–30 cm depth of soil, bulk density was 0.98 g·cm^−3^, total porosity was 20.47%, water content was16.42%, alkali-hydrolyzable nitrogen was 32.58 mg·kg^−1^, available potassium was 180.98 mg·kg^−1^, available phosphorus was 19.63 mg·kg^−1^, pH was 7.97). We designed a two-factor randomized block experiment (one factor is soil metham-sodium fumigation that causes chemical hazard, and the other one is microbial fertilizer supplement, which is used to restore and improve soil health), which included four four treatments: CK, no treatment; SFM, metham-sodium fumigation (metham-sodium was produced by Shandong Yulai Chemical Technology Co., Ltd., Linyi, China); MF, Special 8™ microbial fertilizer supplement (Special 8™ was produced by Qingdao Yuanhui Biological Environmental Protection Technology Co., Ltd., Qingdao, China); MMF, fumigation and microbial fertilizer supplement; three replicates were set for each treatment, and the plot area was 20 m^2^ (10 m × 2 m).

Firstly, we sprayed metham-sodium solution (the concentration is 42%, 375 Kg/ha, with 80 times dilution by water) in ditches (25~30 cm width and 15~20 cm depth), then covered the ditches with soil and plastic film immediately. After 15 days of fumigation, the plastic film was unwrapped, and soil was aired for 5 days. Finally, we supplied special 8™ microbial fertilizer (67.5 L/ha) in ditches and sowed the lily bulb seed. The lily cultivar was sowed with a density of 0.30 m × 0.15 m per plantlet in 18 April 2019 (the bulb seed was approximately 17 ± 2 g) and managed with the same agronomic management and fertilization regime based on traditional agricultural practices in this region with an organic fertilizer (11250 Kg/ha) and without irrigation.

Soil samples were collected at the seedling flowering stage (28 July 2019). The 5-point sampling method was selected for each plot, 4 plants for each point, and a total of 20 plants were selected within each plot. The rhizosphere soil samples were obtained from the soil adhering to plants roots (plants were gently shaken by hand). The rhizosphere soils from the 20 plants were mixed to generate one soil sample; all 16 soil samples were collected in this same manner. Each soil sample was divided into two subsamples: one was brought to the laboratory on dry ice and stored at −80 °C for downstream applications (DNA extraction). The remainder of the sample was air-dried for soil characteristic detecting (pH, EC organic matter, available nitrogen, available phosphorus, available potassium), and the data were described in our published work [18].

### 2.2. Determination of Plant Growth and Yield

After the soil samples were collected from the plant roots, 20 plants for each treatment were brought to the laboratory to determine the seedling indexes for plant growth status evaluation. The seedling index was calculated using the formula described in our previous study [22]: Seedling indexes = (Shoot diameter/Plant height + Dry biomass underground parts/Dry biomass aboveground parts) × Plant dry weight. In addition, we evaluated the mother bulb weight from 20 plant per plot after it had grown for 1 year in 3 October 2019. And after 3 years of growing, we harvested the lily and tested the bulbs yields for each plot in 8 October 2021.

### 2.3. High-Throughput Sequencing by Illumina NovaSeq

#### 2.3.1. DNA Extractions

DNA from different samples was extracted using the E.Z.N.A. ^®^Soil DNA Kit (D4015, Omega, Inc., Norcross, GA, USA) according to manufacturer’s instructions. The reagent, which was designed to uncover DNA from trace amounts of sample, was shown to be effective for the preparation of DNA of most bacteria. Nuclear-free water was used for blank. The total DNA was eluted in 50 μL of Elution buffer and stored at −80 °C until measurement in the PCR by LC-Bio Technology Co., Ltd., Hang Zhou, China.

#### 2.3.2. PCR Amplification, ITS Sequencing and 16S rDNA Sequencing

The ITS2 region of the fungi small-subunit rRNA gene was amplified with slightly modified versions of primers ITS1-F (5′-GTGARTCATCGAATCTTTG-3′) ITS2- R(5′-TCCTCCGCTTATTGATATGC-3′) [23]. The 16S region of bacteriasmall-subunit rRNA gene was amplified with slightly modified versions of primers 341F (5′-CCTACGGGNGGCWGCAG-3′) 805R (5′-GACTACHVGGGTATCTAATCC-3′) [24]. The 5′ ends of the primers were tagged with specific barcodes per sample and sequencing universal primers. PCR amplification was performed in a total volume of 25 μL reaction mixture containing 25 ng of template DNA, 12.5 μL PCR Premix, 2.5 μL of each primer, and PCR-grade water to adjust the volume. The PCR conditions to amplify consisted of an initial denaturation at 98 ℃ for 30 s; 32 cycles of denaturation at 98 °C for 10 s, annealing at 54 °C for 30 s, and extension at 72 °C for 45 s; and then final extension at 72 ℃ for 10 min.

The PCR products were confirmed with 2% agarose gel electrophoresis. Throughout the DNA extraction process, ultrapure water, instead of a sample solution, was used to exclude the possibility of false-positive PCR results as a negative control. The PCR products were purified by AMPure XT beads (Beckman Coulter Genomics, Danvers, MA, USA) and quantified by Qubit (Invitrogen, Carlsbad, CA, USA). The amplicon pools were prepared for sequencing and the size and quantity of the amplicon library were assessed on Agilent 2100 Bioanalyzer (Agilent, Palo Alto, CA, USA) and with the Library Quantification Kit for Illumina (Kapa Biosciences, Woburn, MA, USA), respectively. The libraries were sequenced on NovaSeq PE250 platform.

#### 2.3.3. Data Analysis

Samples were sequenced on an Illumina NovaSeq platform according to the manufacturer’s recommendations, provided by LC-Bio. Paired-end reads were assigned to samples based on their unique barcode and truncated by cutting off the barcode and primer sequence. Paired-end reads were merged using FLASH. Quality filtering on the raw reads was performed under specific filtering conditions to obtain the high-quality clean tags according to the fqtrim (v0.94). Chimeric sequences were filtered using Vsearch software (v2.3.4). After dereplication using DADA2, we obtained a feature table and feature sequence. Alpha diversity and Beta diversity were calculated by being normalized to the same sequences randomly. Then, according to SILVA (release 132) classifier, feature abundance was normalized using the relative abundance of each sample. Alpha diversity was applied in analyzing the complexity of species diversity for a sample through four indices, including Chao1, Observed species, Goods coverage, and Shannon [25], and all those indices in our samples were calculated with QIIME2 [26]. Beta diversity was calculated by QIIME2, and the graphs were drawn by R package, Blast was used for sequence alignment, the bacterial feature sequences were annotated with SILVA database [27] for each representative sequence, and the fungal feature sequences were annotated with RDP [28] and unite database [29].

To examine the statistical significance of structural similarity among communities across different treatments, Principal Component Analysis (PCA) applying Euclidean distance metrics [30] and hierarchical clustering based on the UPGMA distance matrix [6] was performed using the R package. To show the evolutionary relationship among microorganism communities, the phylogeny tree was performed by R package. To compare community characteristics in greater detail, Venn diagrams at OTUs level were created with the R package. To determine the significantly different important microbial taxa, the ANOVA (analysis of variance) was carried out. To further assess the relationship between the microorganism community composition and physicochemical properties in different soil treatments, a redundancy analysis (RDA) was performed based on fungal and bacterial abundance at the genus levels and physicochemical parameters using the R package.

#### 2.3.4. Accession Numbers

The sequences obtained in this study were submitted to the NCBI Sequence Read Archive (SRA) under the Bioproject ID PRJNA805390.

## 3. Results

### 3.1. The Plant Growth and the Lily Yield Analysis under Different Treatments

Metham-sodium soil fumigation had a negative influence on the plants, which significantly decreased the plant growth status, while microbial fertilization significantly improved bulb growth when compared to chemical fumigation. In the first year after the treatments, compared to CK, SFM, and MMF, the seedling index in MF significantly decreased and improved. In addition, the negative influence of soil fumigation and the positive influence of microbial fertilization lasted for a long time: After 3 years of cropping, SFM treatment decreased bulb yield by 12.32% compared to CK control, and MMF treatment improved yield by 5.59% compared to SFM treatment. (Table 1).

### 3.2. Quantity of the Microorganism Diversity in Rhizosphere Soil

Sequencing of the fungal ITS regions of 28S rRNA genes resulted in a total of 511,074 high quality, chimera-free reads. Among the reads, 114,317, 163,431, 109,778, and 123,548 sequences were obtained for each of the CK, MF, MMF, and SFM soil samples, respectively. The OTU distribution Venn analysis demonstrated that there were a total of 1508 OTUs, and 226, 407, 245, and 137 unique OTUs in CK, MF, MMF, and SFM soil samples, respectively (Appendix A). Sequencing of the bacterial V3–V4 hypervariable regions of 16S rRNA genes resulted in a total of 364,067 high quality, chimera-free reads. Among the reads, 112,071; 96,912; 52,933; and 102,151 sequences were obtained for each of the CK, MF, MMF, and SFM soil samples, respectively. The OTU distribution Venn analysis demonstrated that there were a total of 9561 OTUs and 2069, 964, 1692, and 1785 unique OTUs in CK, MF, MMF, and SFM, respectively (Appendix A).

Both the Shannon analysis (Appendix A) and Good’s coverage analysis (the Good’s coverage value 96–100% were used in this analysis) also indicated that the sequence amount was enough to represent the true microorganism in the sample, and deeper sequencing identification was successful.

### 3.3. Overall Structural Changes of Microorganism Communities in Rhizosphere Soil

Alpha diversity indices were evaluated based on OTUs. For fungi, the Chao1 richness (Figure 1A) and Shannon’s diversity (Figure 1B) significant differed among all treatments, while the observed species and PD whole tree did not significantly change. When compared to CK, under FM treatment, Chao1 richness and Shannon’s diversity increased by 6.70% and 35.09%, respectively. When compared to SFM treatment, under MMF treatment, Chao1 and Shannon indices increased by 84.08% and 8.76%, respectively. As for bacteria, Alpha diversity indices did not significantly shift among all these treatments (Figure 1).

Beta diversity was evaluated by PCA based on all the microorganism OTUs (both fungal OTUs and bacterial OTUs together). The result was partially successful in representing sample data. The replicates of each sample were found to group together except in samples SMF3 and MMF2. Moreover, these four groups were clearly separated into three clusters: CK microorganism community, MF microorganism community, as well as SFM and MMF microorganism community. The first principle component axis (PCA1), which contributed 53.29% of the total variation, and the second component axis (PCA2), which contributed 11.96% of the variation, explained 65.25% of the variation (Figure 2).

### 3.4. Taxonomic Distributions of Microorganism Enriched in the Replant Soil

A total of seven fungal phyla were identified, in which the dominant phyla were *Ascomycota, Basidiomycota*, *Zygomycot*a, and *Chytridiomycota*, with average RA values of 83.76%, 8.66%, 2.01%, and 1.18%, respectively. In addition, at the class levels, 24 classes were identified. The most dominant classes were *Sordariomycetes, Pezizomycetes, Eurotiomycetes,* and *Agaricomycetes* (average RA > 5%). A total of 275 genera were detected; among them, 22 genera were dominant (average RA > 1%), and the most accumulated genera (average RA > 5%) were *Gibberella, Gliocladium, and Kotlabaea,* in which the average RA were 16.10%, 15.83%, and 5.83%, respectively. At the species levels, 316 species were detected. Among them, nine species were dominant (average RA > 1%), and the most accumulated species (average RA > 5%) were *Acremonium nepalense* and *Fusarium tricinctum*. A total of 35 bacterial phyla were identified, in which the dominant phyla were *Proteobacteria, Actinobacteria, Acidobacteria, Gemmatimonadetes, Chloroflexi, Planctomycetes, Bacteroidetes, Verrucomicrobia,* and *Nitrospirae*, in which the average RA was 26.56%, 20.52%, 15.62%, 13.46%, 7.35%, 2.33%, 1.50%, 1.26%, and 1.02%, respectively, and the phyla P*roteobacteria* accounted for 30.69% and 30.98% of MF and MMF, while *Actinobacteria* accounted for 23.75% and 23.30% of CK and SFM. At the genus levels, 447 genera were identified, and among them, 18 genera were dominant (average RA > 1%), and the most accumulated genera (average RA > 5%) were *Gemmatimonas, Sphingomonas* and MND1, with an average RA of 9.41%, 7.40%, and 5.22%, respectively. At the species levels, 415 bacterial species were detected, and among them, 15 species were dominant (average RA > 1%), and the most accumulated species (average RA > 5%) were uncultured *Gemmatimonas* sp. And uncultured *Sphingomonas* sp (Figure 3B). The phylogeny trees showed the evolutionary relationships among fungal genera, as well as bacterial genera (Appendix A).

### 3.5. Special Microorganism Community Structure Dynamics under Different Treatments

The fungal and bacterial community structures were significantly different among four treatment soils. Some dominant phyla (average RA > 5%) significantly shifted, such as the fungal phyla *Ascomycota* and *Basidiomycota* (*p* < 0.05), and the bacterial phyla *Pro*t*eobacteria, Gemmatimonadetes,* and *Chloroflexi* (*p* < 0.05). Among them, in the best fertilizer treatment MF, the highest abundance phyla were *Basidiomycota* and *Proteobacteria*, while the lowest abundance phyla were *Ascomycota* and *Gemmatimonadetes* (Table 2 and Table 3).

We listed the dominant members (average RA > 0.1%) whose proportion significantly shifted (*p* < 0.05). At genus levels, in the best fertilizer treatment MF, the highest abundance groups were the genera *Mortierella* and *Sphingomonas*, and the lowest abundance genera were *Isaria* and *Gemmatimonas* (Table 2 and Table 3). As for fungi, under the microbial fertilizer supplement treatments (MF and MMF treatments), 40.00% genera enriched, and 6.67% genera decreased significantly when compared to no microbial fertilizer supplement treatments (CK and SFM treatments). Under the MF treatment, 46.67% genera enriched significantly and 33.33% genera decreased significantly when compared to CK, and under the MMF treatment, 53.33% genera enriched significantly and 6.67% genera decreased significantly when compared to SFM treatment. As for bacteria, under the microbial fertilizer supplement treatments (MF and MMF treatments), 61.54% genera enriched and 23.08% genera decreased significantly when compared to no microbial fertilizer supplement treatments (CK and SFM treatments). Compared with CK, under MF treatment, 53.85% genera enriched and 30.77% genera decreased significantly, and compared with SFM, under the MMF treatment, 61.54% genera enriched and 30.77% genera decreased significantly. Thus, it seemed that the microbial fertilizer supplement absolutely drove more dominant fungal and bacterial member accumulation both in untreated soil and the fumigated soil (Table 2 and Table 3).

### 3.6. RDA Analysis of Soil Physical and Chemical Properties and Microorganism under Different Treatments

Soil physicochemical properties significantly differed among the four treatments (Appendix A). The microorganism RDA model (including fungal and bacterial communities) showed that the RDA1 and RDA2 explained 39.38% of the total variance, and all the soil physicochemical character indexes made contributions to the microorganism community structure rebuilding, among them, available potassium and organic matter played more influential roles in microorganism shaping. The RDA plot of the microorganism community composition also clearly showed that the samples in CK, SFM, MF, and MMF (except the soil sample MF3) were significantly different. (Figure 4).

## 4. Discussion

### 4.1. Microorganism Diversity in Different Soil Treatments

The study revealed that metham-sodium soil fumigation hazard significantly decreased plant growth status and decreased bulb yield, as well as it the influenced soil microorganism community impressively, but the microbial fertilization improves soil health when compared to chemical fumigation in Sweet Lily.

Microorganism PCA showed that these four groups were clearly separated into three clusters: CK microorganism community, MF microorganism community, as well as SFM and MMF microorganism community (Figure 2). These results indicated that the microbial fertilizers influenced the microorganism structure shaping in lily rhizosphere soil. RDA also revealed that the physical and chemical properties of soil influenced the microorganism community reconstruction (Figure 4).

In nature, soil biodiversity has a positive correlation with the productivity and sustainability of a system [31], and the loss and simplification of soil community composition impair multiple ecosystem functions, including plant diversity, decomposition, nutrient retention, and nutrient cycling [32]. Microorganism diversity is an important part of soil biodiversity. Our previous research showed that long-term replanting decreased both soil fungal diversity and abundance in Lanzhou lily replanting system, which was one reason for soil degradation in this replanting system [15]. In this study, the fungal abundance and diversity decreased in SFM (Figure 1), which meant that the Alpha diversity loss was an important mechanism for the soil degradation, which resulted from the chemical hazards. On the other side, the Alpha diversity increased in MF treatment (Figure 1), and it was helpful to alleviate CRPs, as well as restore and promote soil health. The special microorganism community structure dynamics analysis also showed that metham-sodium fumigation reduced more dominant fungal than the bacteria, and the microbial fertilizer supplement drove more dominant fungal and bacterial genera increase both in untreated soil and fumigated soil (Table 2 and Table 3).

Our previously study showed that in the lily 0–9 replanting system, in L0 ((non-replant lily) and L3 (crop lily for 3 years) treatments, more than 99% of fungal groups could not be identified. However, after 6 years of replanting, the soil contained few unknown fungal members; in L6 and L9 treatments (monocrop lily for 6, 9 years, respectively), there were 0.57% and 1.24% sequences that remained unclassified [15]. In this study, the experimental field replanted Lanzhou lily for 9 years. In CK and SFM treatments, 0.35% and 0.32% fungal sequence cannot be identified, but in FM and MMF treatments, the unidentified fungal sequence increased to 8.47% and 5.08% (Figure 3). This result means the microbial fertilizer supplement results in a part of unknown specific fungi appearance. We believe these unknown specific fungi must make a great contribution to soil productivity and increase the fungal Alpha diversity.

Recently, one important literature reviewed that fumigation can decrease bacterial abundance and diversity, and these decreases appear transient and tend to diminish or disappear after 4 weeks. Increases in bacterial diversity and abundance can occur after fumigation, but are less common [19]. In addition, fumigation with metam-sodium has been reported to reduce bacterial diversity after 30 days [33], 63 days [34] and 112 days [35] compared to non-treated soils. In this study, the bacterial abundance and diversity were the same after 100 days of metam-sodium fumigation compared to non-treated soils. Our results supported these previous conclusions. This is because the plant species and their roots exudate with soil microorganisms [36], as well as the temperature and moisture [18] can influence fumigant degradation. In addition, we believed that the bacterial abundance and diversity must decreas firstly, then recover to the level in non-treated soils. Compared to the bacteria, the data on the effect of fumigants on fungi communities is limited. One literature reported that gradual increases in fungal diversity were observed over a 112-day incubation period in soils treated with Allyl isothiocyanate [20], which is similar to the results in this study.

### 4.2. The Possible Fungal Groups Related to the Alleviation of Lily CRPs and Soil Health

In this study, we detected the genus *Mortierella* whose proportion was dominant and significantly increased by more than 100 times in microbiology fertilizer treatments when compared to non-microbiology fertilizer-treated soils (in CK, SFW, FM, MMF, RA = 0.03%, 0.04%, 3.04%,and 4.92%, respectively, Table 2). Scientists suggests that some fungal species may utilize Allyl isothiocyanate as a C and/or N source and colonize empty niches after fumigation [19]. For example, the fungal genus *mortierella* can degrade different fumigants including Allyl isothiocyanate [37]. Both Allyl isothiocyanate and metam-sodium are broad-spectrum fumigants with fungicidal and nematocidal properties, and they have similar chemical structure and element composition (the Chemical formula: C_4_H_5_NS and C_2_H_4_NNaS, respectively). Therefore, the result suggests that *Mortierella* may make a great contribution in recovering or promoting the fumigation effect. Microbiology fertilizer supplement induced *Mortierella* sp. accumulation, which enhances these fungal species to use fumigant as a C and/or N source and proliferation, and it is helpful to promoting the fumigation effect.

In this study, we detected one putative harmful fungal genus *Cryptococcus,* this genus significantly enriched with microbial fertilizer supplement (Table 2), so we believed that these fungal members might be related to CRPs incidence (Table 2). The genera *Cryptococcus* had high abundances and negative correlations with plant growth in apple replant orchards [7]. These results partially supported our previous finding that *Cryptococcus was* associated with Lanzhou CRPs incidence [15]. *Fusarium* ssp. are well-known fungal pathogens in crop soil-borne diseases. Our previous study demonstrated that the accumulation of some *Fusarium* ssp. played an important role in the lily CRPs. Our team also identified *F. oxysporum*, *F. tricinctum,* and *F.solani* as pathogens of Lanzhou lily wilt disease by using an in-vitro culture method [38]. As well as other Fusarium species were reported to cause lily wilt disease [39,40]. In this study, *F. oxysporum* was significantly increased in SFM compared with CK (Table 2), which might explain why *F. oxysporum* was easy to colonize in relatively clean soil because of the soil fumigation. On the other side, *F. oxysporum* was significantly decreased in MMF compared with SFM; it might be explained that fertilizer produce antagonism to control this pathogen. In this research, we used Special 8™ microbial fertilizer, which can produce antagonism to the pathogens. Special 8™ microbial fertilizer was mainly composed of the beneficial bacterial strains of *Pseudomona*s, *Arthrobacter,* and *Lactobacillus*, as well as 19 other beneficial bacteria with a total of 15,000 cfu/g microorganisms. *Pseudomonas* and *Arthrobacter* can inhibit the occurrence of soil-borne plant diseases [8] and induce the disease-resistant soil formation in continuous cropping cultivation [41]. We also detected *Pseudomonas* sp. (average RA = 0.10%) and *Arthrobacter* (average RA = 0.80%). Moreover, as we mentioned above, MF treatment induced *Sphingomona*s significant accumulation, and *Sphingomonas* is an important antagonistic bacterium [24]. All the analysis was helpful to explain how the microbial fertilizers produce antagonism that reduces harmful microorganism and regulates microorganism structures. In this study, we identified other *Fusarium* ssp., including lily wilt disease pathogens *F. tricinctum* and *F.solani*. However, all these species did not significantly shift (Appendix A), moreover, some *Fusarium* species that function in soil health maintenance were not reported, so we believed it is necessary to explore this aspect.

### 4.3. The Possible Bacterial Groups Related to the Alleviation of Lily CRPs and Soil Health

To our excitement, in this study, we found out that the bacterial phylum *Proteobacteria* and its member *Sphingomonas*
*are* involved in soil health recovery and promotion, which could be regarded as soil health indicators in Lanzhou Lily continuous cropping system as we previously reported [14]. These two groups significantly enriched with microbial fertilizer supplement (Table 3). Our team demonstrated that *Proteobacteria* and *Sphingomonas* are indicators of soil health in 0–9 years of Lanzhou lily replanting system. Thus, this result supported our previous conclusions. *Proteobacteria* are generally reported to be the most dominant bacteria in soil and include many members classified as plant growth-promoting bacteria [7,42], which contribute to nutrient acquisition, provide protection against disease, and are positively associated with the incidence of bacterial disease [8]. *Sphingomonas* is an important plant growth promoter [8] that has biocontrol effects on pathogenic bacteria [43]. *Sphingomonas* is also associated with the incidence of apple replant disease (ARD) [6,7]. RDA model also supported that *Sphingomonas* was an important member involved in soil health recovery and promotion. In RDA model, *Sphingomonas* negatively correlated to the bacterial genus *Gemmatimonas*, as well as fungal genera *Gemmatimonas, Exophiala*, *Fusarium,* and *Gibberella. Gemmatimonas* was a putatively harmful genus, it was associated to CRPs [6,7,8,9], and it significantly depleted with microbial fertilizer supplement in this study (Table 3), so we believed that these genera *Gemmatimonas* might be related to soil health degradation. *Exophiala* had high abundances and negative correlations with plant growth in apple replant orchards [7]. Our team previously found that *Fusarium, Gibberella,* and *Alternaria* associated to Lanzhou CRPs incidence [15]. In this study, we identified some *Sphingomonas* species, but there are 89.97% species uncultured or unidentified (Appendix A). Therefore, there is important theoretical and application value in looking for some functional *Sphingomonas* strains by in-vitro culture technology, and there are a lot of gaps from this aspect.

## 5. Conclusions

The study revealed that metham-sodium soil fumigation hazard significantly decreased plant growth status and bulb yield, as well as changed the microorganism structures impressively. Under chemical fumigation conditions, fungal diversity loss and overall microorganism reduction, which impair multiple ecosystem function, in conjunction with the increase of harmful microorganism species such as genera *Gemmatimonas and Cryptococcus,* and the pathogenic fungal species *Fusarium oxysporum,* are the main reasons for soil degradation. On the other hand, with a microbial fertilizer supplement, the fungal diversity increasde, and a part of some unidentified fungi appearance, as well as these beneficial microorganism groups’ (the bacterial phylum *Proteobacteria*, genus *Sphingomonas,* and fungal genera *Mortierella*) accumulation, together with the harmful microorganism group depletion, played important roles in restoring and improving soil health. The bacterial phylum *Proteobacteria* and its member genus *Sphingomonas* are soil health indicators in the lily replanting system, and they are involved in soil health recovery and promotion under the chemical fumigant condition. More to the point, microbial fertilizer is safe for human health. The results also emphasized that whether soil is chemically fumigated or not, bio fertilizer is effective in keeping a soil microorganism’s community in balance and ensuring soil productivity.

## Figures and Tables

**Figure 1 jof-08-00847-f001:**
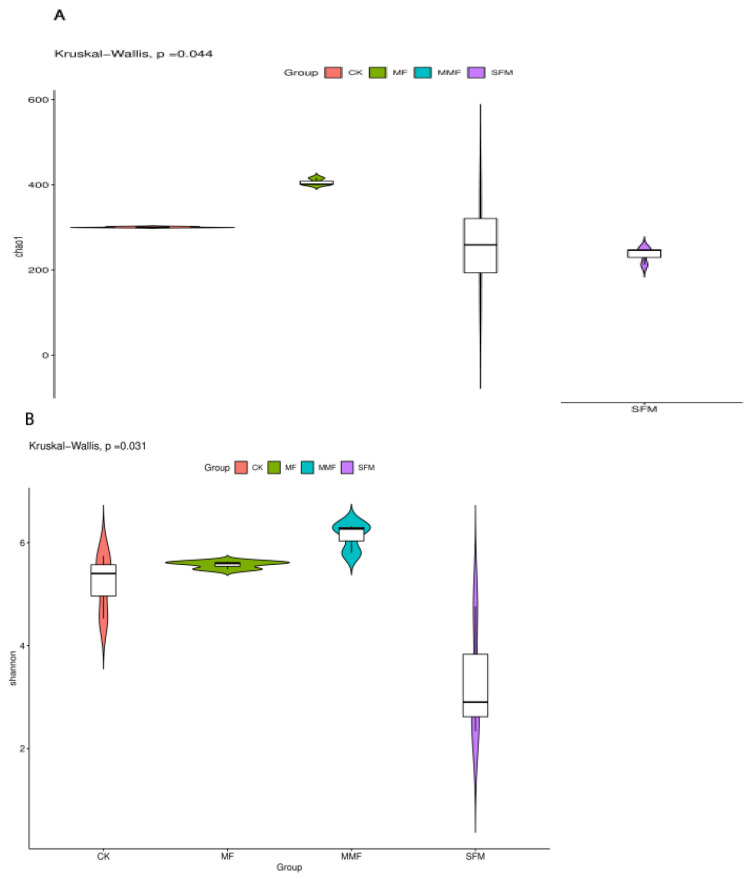
Fungal Chao1 richness (**A**) and Shannon index (**B**) of Lanzhou lily rhizosphere soil among different treatments from Illumina NovaSeq data.

**Figure 2 jof-08-00847-f002:**
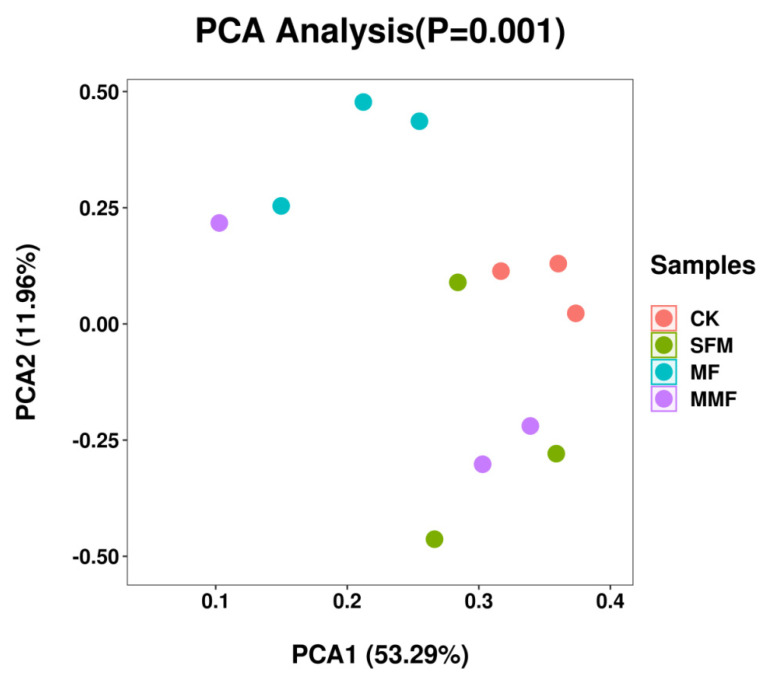
Principal Component Analysis (PCA) showing fungal and bacterial overall structural changes of Lanzhou lily rhizosphere soil among different treatments from Illumina NovaSeq.

**Figure 3 jof-08-00847-f003:**
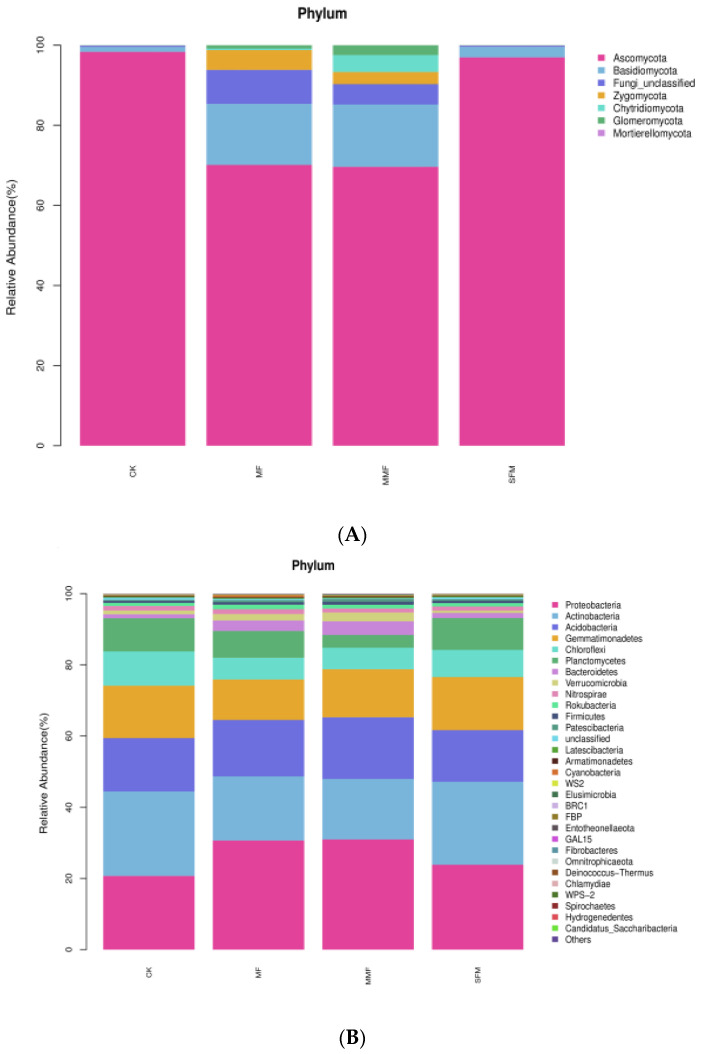
The taxonomic distribution of fungi and bacteria of Lanzhou lily rhizosphere soil among different treatments from Illumina NovaSeq: top 7 fungal phyla (**A**) and top 30 bacterial phyla (**B**).

**Figure 4 jof-08-00847-f004:**
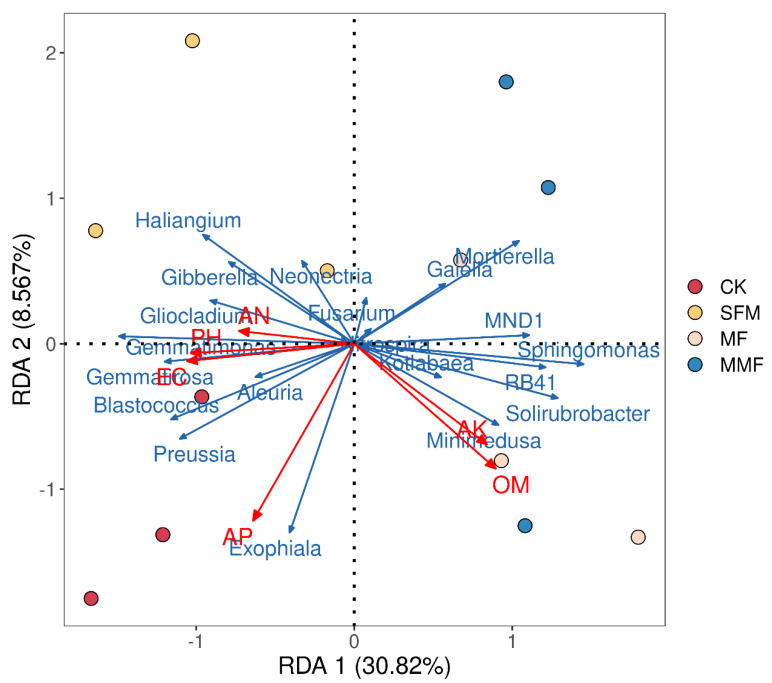
Redundancy analysis (RDA) of microorganism communities and soils physicochemical properties: microorganism included top 10 fungal genera and top 10 bacterial genera; OM, EC, AK, AN, and AP represented organic matter, electrical conductivity, available potassium, alkali-hydrolyzable nitrogen, and available phosphorus, respectively.

**Table 1 jof-08-00847-t001:** The lily growth status of Lanzhou lily rhizosphere soil among different treatments: The seedling index was calculated at lily flowering stage in July 2019; Mother bulb weight means the seed bulb weight after it had grown for 1 year, and it was evaluated in October 2019; Bulb yield referred to the bulb weight after it had grown for 3 years, and it was evaluated from all the plants in each plot in October 2021: the significance is 0.05 for lower letters (a, b, c) presented in table.

Treatments	Seedling Index	Mother Bulb Weight (g)	Bulb Yield (kg/h m^2^)
CK	16.75 ± 0.98 b	29.95 ± 1.04 ab	19,550.83 ab
SFM	18.88 ± 2.19 b	24.34 ± 0.72 c	17141.47 c
MF	27.28 ± 0.62 a	33.91 ± 1.46 a	22,103.00 a
MMF	16.15 ± 1.24 b	27.76 ± 0.48 b	18,099.97 b

**Table 2 jof-08-00847-t002:** The fungal group significantly changed (*p* < 0.05) total reads of Lanzhou lily rhizosphere soil from Illumina NovaSeq data (the phyla with average RA > 5%, and the genera with average relative abundance > 0.1% and the species with average relative abundance > 0.05%; the accession number is ID PRJNA805390): the significance is 0.05 for lower letters (a, b, c) presented in table.

Treatments	CK	SFM	MF	MMF
(a) enriched phyla and genera and species with microbial fertilizer supplement				
*p__Basidiomycota*	1.21% b	2.64% b	15.25% a	15.55% a
*g__Mortierella*	0.03% c	0.04% c	4.92% a	3.04% b
*g__Funneliformis*	0.00% c	0.02% c	0.42% b	2.38% a
*g__Spizellomyces*	0.03% c	0.02% c	0.11% b	2.48% a
*g__Cryptococcus*	0.05% c	0.03% c	0.59% b	1.78% a
*g__Rhizophydium*	0.00% b	0.00% b	0.01% b	1.09% a
*g__Ceratobasidium*	0.01% c	0.00% c	0.27% b	0.47% a
*s__Mortierella_alpina*	0.02% c	0.03% c	1.69% a	1.14% b
*s__Cryptococcus_aerius*	0.03% c	0.02% c	0.40% b	0.94% a
*s__Spizellomyces_acuminatus*	0.00% b	0.00% b	0.03% b	1.16% a
*s__Mortierella_elongata*	0.00% c	0.00% c	0.58% a	0.42% ab
*s__Cryptococcus_magnus*	0.00% c	0.00% c	0.05% b	0.74% a
*s__Ceratobasidium_sp_AG_I*	0.00% b	0.00% b	0.01% b	0.34% a
(b) depleted phyla and genera and species with microbial fertilizer supplement				
*p__Ascomycota*	98.35% a	96.95% a	70.11% b	69.64% b
*g__Isaria*	1.25% a	0.08% b	0.00% c	0.00% c
*s__Isaria_cicadae*	1.25% a	0.08% b	0.00% c	0.00% c
(c) enriched or depleted phyla and genera and species with microbial fertilizer supplement				
*g__Kotlabaea*	9.01% b	0.65% c	12.84% a	0.82% c
*g__Ochroconis*	1.08% a	0.13% b	1.28% a	0.09% b
*g__Bradymyces*	0.62% a	0.01% c	0.26% b	0.06% c
*g__Phialophora*	0.77% a	0.00% b	0.01% b	0.00% b
*g__Pochonia*	0.00% c	0.05% b	0.04% b	0.45% a
*g__Fusariella*	0.38% a	0.02% c	0.14% b	0.00% c
*g__Conocybe*	0.01% c	0.07% bc	0.04% bc	0.41% a
*g__Typhula*	0.32% a	0.00% c	0.18% b	0.00% c
*s__Kotlabaea_unclassified*	9.01% b	0.65% c	12.84% a	0.82% c
*s__Coprinellus_bisporus*	0.00% c	1.24% b	2.46% a	0.00% c
*s__Cylindrocarpon_liriodendri*	0.03% c	0.10% b	0.05% bc	2.98% a
*s__Ochroconis_humicola*	1.03% a	0.11% b	1.25% a	0.07% c
*s__Phialophora_europaea*	0.77% a	0.00% b	0.01% b	0.00% b
*s__Conocybe_rickenii*	0.01% c	0.07% b	0.04% bc	0.41% a
*s__Typhula_maritima*	0.32% a	0.00% c	0.18% b	0.00% c
*s__Fusarium_oxysporum_f_sp_psidii*	0.00% c	0.24% a	0.00% c	0.08% b

**Table 3 jof-08-00847-t003:** The bacterial group significantly changed (*p* < 0.05) total reads of Lanzhou lily rhizosphere soil from Illumina NovaSeq data (the phyla with average RA > 5%, and the genera with average relative abundance > 0.1% and the species with average relative abundance > 0.05%; the accession number is ID PRJNA805390): the significance is 0.05 for lower letters (a, b, c) presented in table.

Treatments	CK	SFM	MF	MMF
(a) enriched phyla and genera and species with microbial fertilizer supplement				
*p__Proteobacteria*	20.71% b	23.87% b	30.69% a	30.98% a
*g__Sphingomonas*	1.70% bc	2.10% b	4.36% a	4.67% a
*g__Altererythrobacter*	0.48% b	0.32% c	0.36% bc	0.69% a
*g__Ellin6055*	0.16% b	0.25% b	0.57% a	0.57% a
*g__Luteimonas*	0.14% c	0.19% c	0.80% a	0.40% b
*g__Candidatus_Udaeobacter*	0.16% c	0.10% c	0.45% ab	0.57% a
*g__Stenotrophobacter*	0.16% bc	0.14% c	0.19% b	0.28% a
*g__Terrimonas*	0.04% b	0.05% b	0.13% a	0.16% a
*g__Ellin6067*	0.35% b	0.49% b	0.81% a	0.87% a
*s__uncultured_Luteimonas_sp*	0.10% c	0.03% c	0.66% a	0.29% b
(b) depleted phyla and genera and species with microbial fertilizer supplement				
*p__Gemmatimonadetes*	9.63% a	7.64% b	6.09% bc	6.05% bc
*p__Chloroflexi*	9.63% a	7.64% b	6.09% c	6.05% c
*g__Gemmatimonas*	4.57% a	4.51% a	3.54% b	3.70% b
*g__Gemmatirosa*	1.10% a	0.98% ab	0.78% b	0.51% c
*g__OM27_clade*	0.27% a	0.22% a	0.09% c	0.17% bc
*s__uncultured_Pseudonocardia_sp.*	0.24% ab	0.31% a	0.00% c	0.11% b
*s__uncultured_Virgisporangium_sp.*	0.10% a	0.09% a	0.01% b	0.04% b
(c) enriched or depleted phyla and genera and species with microbial fertilizer supplement				
*g__Candidatus_Alysiosphaera*	0.22% a	0.14% b	0.12% c	0.18% ab
*g__Hirschia*	0.09% c	0.23% a	0.16% b	0.11% bc

## Data Availability

The sequences obtained in this study were submitted to the NCBI Sequence Read Archive (SRA) under the Bioproject ID PRJNA805390.

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
