# Peer review of "Microbial Fertilization Improves Soil Health When Compared to Chemical Fumigation in Sweet Lily"

_jof, 2022, doi:10.3390/jof8080847_

Round 1
Reviewer 1 Report
This is an interesting manuscript about the determination of the bacterial and fungal community structure and some specific microbial groups in the rhizosphere soil of sweet lily, as known in China, after metham-sodium soil fumigation and microbial fertilizer supplement by using a metagenomic analysis of the treated soils.
A great deal of work was conducted here. The manuscript is written well, and very accurate. The results obtained were valuable and persuasive. I would therefore recommend accepting this manuscript.
Reviewer 2 Report
Title Microbial fertilization improves soil health when compared to 2 chemical fumigation in Sweet Lily
The current manuscript has a good idea but needs some modifieds
Abstract:
-it is good, but the authors should consider the proposed changes for improving the clarity of the content. Such comparison among different treatments and suggesting microbial fertilization because it is safe for human health and chipper etc...
Keyword: show good
-Introduction part is appropriate but a few things are needed for further improvements especially the study aims should be added and rewritten sentences" In this paper, we will report how the soil fumigation and microbial fertilizer changed the microorganism and which the specific microorganism members involved in soil health damage, recovery and promotion.
Add some studies about the study with highlighting research gaps, There is only one reference in 2021 and the other is older, it needs to update the references
Materials and methods:
-this part describes very well by using suitable subheadings. However, it needs few modifications and details of selecting primers and amplification conditions in the revised version to enhance clarity.
-Author said: The 16S region of the eukaryotic (bac- 165 teria) small-subunit rRNA gene was amplified with slightly modified versions of primers 166 341F (5'-CCTACGGGNGGCWGCAG-3') 805R (5'-GACTACHVGGGTATCTAATCC- 167 3')[20].
-In lone 166, 167, what is the meaning of eukaryotic bacteria???
-And PCR program for ITS or for 16S rDNA amplification, author said annealing at 54 oC
-Data analysis: need to add references of each analysis program.
Results
-Results part needs to combine and needs minor revision and it needs some figs of sequencing analysis of obtained taxa in plants rhizosphere. And phylogeny tree of fungi and bacterial strains.
- refer to the most species that are found in the best fertilizer treatment of the four treatments and the lowest one.
- accession number of study should be added in the table or separated from A to Z
- Discussion
Appear as a good part but need newer references
Conclusion:
-Improve this part with respect to formulated objectives.
- not that the biofertilizer methods are the best for Agriculture and Human
References:
-Cross-check the references in the text and reference cite.
Reviewer 3 Report
The submitted manuscript entitled “Microbial fertilization improves soil health when compared to chemical fumigation in Sweet Lily” by Hui et al., explored the bacterial and fungal community structure and some specific microbial groups in the lily rhizosphere soil after chemical versus bacterial fertilizer treatments, by using a metagenomic analysis of the treated soils. In this study, the authors attempted to show that metham-sodium soil fumigation (SMF treatment) significantly decreased plant growth, as well as it significantly decreased both soil fungal diversity and abundance at the OTUs levels, while Special 8™ microbial fertilizer supplement (MF treatment) significantly improved plant growth and increased fungal diversity and abundance. However, the bacterial diversity and abundance were not significantly changed between the two treatments. Overall, the authors emphasized that whether soil is chemical fumigated or not, beneficial microorganism supplementary in time is effective on keeping soil microorganism community balance. I find this manuscript acceptable for publications. Please check for the grammatical error, citation format issue, and other minor concerns if any.
Reviewer 4 Report
You have performed some interesting experiments and written up this article to improve the soil used for cultivating the edible sweet Lanzhou Lily. You have gathered a lot of results and have tried to analyze them to obtain a clear indication of a more suitable soil treatment. In a number of places however, the english makes the results and conclusions unclear to me as a reader, and so some of my comments below could be due to misunderstanding of what you are trying to convey
I In the article, you state that Lanzhou Lily has been cultivated as a traditional food crop for some time. What has been the historical method to treat the soil? The SMF method or another? If it is the SMF, then from your results and conclusions, the soil should already be unsuited for Lanzhou lily growth.
2. A clear comparison with the CK control soil should be made throughout, as this is really your baseline measurements. The change conferred by the different treatments on the fungal community should be described in as much detail as the bacterial community changes. As it is, it is difficult to really see the fungal changes brought about by the treatments (especially as you are submitting to the Journal of Fungi).
3. Under the MF and MMF treatment methods, was the actual physical growth of the plant improved, and how can this be related to the microbial soil composition? You mention a significant decrease and improved status for the seedling index (line 218-219). What does this mean?
4. What do you mean when you write ‘reconstructing microorganism structure’ (line 346)? One thing is compositional change, but are there any other structural changes that you have observed. Are there changes in the soil strata microbial composition that you have observed, or what microorganisms, especially at the fungal level, interact with the bulbs?
5. From your results, what would therefore be the optimum soil microbial diversity to ensure productive and long-term Lanzhou Lily cultivations, giving particular response to the particular fungal biodiversity (is an overall increase in fungal numbers good, or in particular species over others)?
6. The whole article needs careful editing and rewriting by either a native English-speaker or a commercial language-checking company. For example, as it is written, I cannot understand what you mean in lines 249-251
7. AP and OM abbreviations need to be spelt out in the main article (line328) not needing to look at Supplementary Table 2 to decipher it. Additionally, the layout of Supplementary Table 2 needs to be corrected. Possibly landscape orientation would make it easier to read?
Round 2
Reviewer 4 Report
I congratulate the authors for making these significant changes to their manuscript, highlighting the impact on the funagl community
